# The Significance of the Microenvironment in T/Nk-Cell Neoplasms

**DOI:** 10.3390/ijms262211225

**Published:** 2025-11-20

**Authors:** Ivan Petković, Michele Ritucci, Ana Stojković, Slavica Stojnev, Aleksandar Popović, Irena Conić, Milica Radić, Miljana Džunić, Miljan Krstić

**Affiliations:** 1Department of Oncology, Medical Faculty Niš, University of Niš, 18000 Niš, Serbia; michelepioritucci@gmail.com (M.R.); irenaconic@yahoo.com (I.C.); milica91nis@ymail.com (M.R.); 2Oncology Clinic, University Clinical Center Niš, 18000 Niš, Serbia; anastojka97@gmail.com (A.S.); popovic992@yahoo.com (A.P.); drmdzunic@gmail.com (M.D.); 3Department of Pathology, Medical Faculty Niš, University of Niš, 18000 Niš, Serbia; slavicastojnev@gmail.com (S.S.); krstic.miljan@gmail.com (M.K.); 4Center of Pathology, University Clinical Center Niš, 18000 Niš, Serbia

**Keywords:** T/NK-cell neoplasms, lymphoma microenvironment, extracellular matrix, cellular stroma, acellular stroma

## Abstract

T/NK-cell neoplasms represent rare and highly diverse cancers, distinguished by variability in their molecular architecture, local inflammatory milieu, and microenvironmental composition, which collectively underpin the diversity of clinical presentations and outcomes. The neoplastic tissue comprises malignant lymphoma/leukemic cells in concert with a spectrum of stromal elements and the acellular extracellular matrix (ECM), collectively constituting the lymphoma microenvironment (LME). These components engage in dynamic, reciprocal interactions, forming a self-regulating ecosystem capable of responding adaptively to both exogenous and endogenous stimuli. Historically, the LME was largely neglected in considerations of lymphomagenesis; however, emerging evidence highlights its pivotal role in driving core oncogenic processes, including sustained proliferative signaling, angiogenesis, immune evasion, and apoptotic resistance. Deciphering the intricate, multidirectional crosstalk among the cellular and acellular constituents of the T/NK-cell neoplastic microenvironment promises to deepen our understanding of disease biology and may inform the development of novel, mechanism-based therapeutic interventions.

## 1. Introduction

T/NK-cell neoplasms constitute a highly heterogeneous and rare group of cancers [1]. Their presumed cells of origin are mature T or NK-cells. It is hypothesized that oncogenic transformation occurs early in the cellular life cycle, yet the malignant clone retains the capacity for terminal differentiation and maturation. This phenomenon likely reflects preservation of the cellular maturation program, in contrast to immature T/NK-cell counterparts. Although these neoplastic cells are phenotypically mature, they harbor extensive molecular abnormalities, which contribute to the marked histological and clinical heterogeneity observed across distinct entities. Clinically, T/NK-cell neoplasms may present as nodal, extranodal, or leukemic disease, depending on the distribution and growth pattern of the malignant cells, although overlapping patterns are frequently observed.

T/NK-cell neoplasms account for approximately 10–15% of all non-Hodgkin lymphomas (NHL) in adults, classifying them as rare entities [1]. A prominent epidemiological feature of these neoplasms is their association with viral infection, notably Epstein–Barr virus (EBV) and human T-cell lymphotropic virus type 1 (HTLV-1), which underlies marked geographic and racial variation in incidence [1]. The occurrence of T/NK-cell neoplasms closely parallels the prevalence of these viral infections in endemic regions, including the Eastern Hemisphere, with Japan, broader Asia, West Africa, and South America being most heavily affected [2,3].

The two leading international consortia have recently issued updated classifications incorporating the most current advances in the field: the 5th Edition of the World Health Organization Classification of Haematolymphoid Tumors (WHO-HAEM5, 2022) and the International Consensus Classification of Mature T/NK-Cell Neoplasms (ICC, 2022) [4,5,6]. A comparative synthesis of these frameworks is presented in Table 1.

Regardless of the classification system employed, a simplified framework commonly categorizes T/NK-cell neoplasms as nodal, extranodal, or leukemic, although overlapping features are frequently observed. Both the WHO-HAEM5 and ICC 2022 have provided integrative approaches to the classification of T/NK-cell neoplasms, incorporating genetics, pathology, immunophenotype, and clinical presentation. A notable advancement in both systems is the unification of nodal T follicular helper (T_FH_) lymphomas into a single entity and the recognition of EBV^+^ nodal T/NK-cell lymphoma as a distinct entity [7].

Given their derivation from a common progenitor, T and NK-cell neoplasms are best considered collectively rather than separately. Mature peripheral T-cells are characterized by membrane CD3 expression, reflecting a fully developed T-cell receptor (TCR). Cytotoxic T-cells are further defined by CD8 expression in conjunction with CD3, along with cytoplasmic cytotoxic granules, including perforin, T-cell restricted intracellular antigen (TIA-1), and granzyme B. In the PTCL-NOS subtype, cytotoxic phenotypes are generally associated with more aggressive clinical behavior, although this is not universal [8,9,10,11]. NK-cells, key effectors in innate immunity that also participate in adaptive responses, consistently express CD16, CD57, and CD56. They typically display only the cytoplasmic CD3 ε chain, while membrane CD3 is absent, a hallmark of ENKTCL [12,13,14]. T-helper cells express membrane CD4 alongside CD3, whereas T_FH_ cells are defined by co-expression of biomarkers such as BCL6, CD10, CXCL13, PD-1, SAP, ICOS, and CCR5 [8,15,16]. Aberrant biomarker expression is a central feature of T/NK-cell neoplastic transformation, while EBV-infected neoplastic cells consistently express latent membrane proteins (LMP1/2) and EBV-encoded RNA (EBER).

The LME constitutes a dynamic, interdependent ecosystem comprising non-malignant cellular components, ECM, and a milieu of cytokines, hormones, and exosomes. Within this ecosystem, neoplastic lymphoma cells are intimately intermixed with stromal and acellular elements, forming a unified neoplastic ensemble. The components of the LME engage in complex, reciprocal interactions, providing both supportive and inhibitory signals to malignant cells. Remarkably, the LME functions as an adaptive endogenous ecosystem, capable of harmonizing its activities in response to continuous exogenous and endogenous influences. Historically underappreciated, the LME is now recognized as a critical regulator of core oncogenic processes, including sustained proliferative signaling, angiogenesis, immune evasion, and apoptosis resistance, and is emerging as a compelling focus for translational research. Advances in understanding these interactions are poised to inform the development of targeted therapeutic strategies.

## 2. The Concept of the Microenvironment in T/NK-Cell Neoplasms

In the classical sense, the LME comprises diverse accessory (non-malignant) cells embedded within acellular matrix components, collectively forming a self-regulating endogenous microecosystem that interacts intimately with malignant lymphoma cells. Functionally, the reactive cellular components, including immune cells, stromal elements, and angiogenic cells associated with the vasculature, together with acellular matrix compounds and the cytokine milieu, establish a complex, dynamic network that modulates signaling pathways, local metabolism, mechanical support, and immune responses, thereby exerting profound influence on lymphoma cell behavior [17]. The LME facilitates malignant cell survival through mechanisms such as sustained proliferative signaling, evasion of growth suppressors, resistance to apoptosis, and immune escape [18]. Figure 1 illustrates the LME as a discrete microenvironmental ensemble composed of its principal cellular and stromal components.

Furthermore, the concept of the matrisome, a functional ensemble of genes encoding ECM and ECM-associated proteins, has been introduced to define the coordinated interactions between cellular and acellular stromal components, highlighting the ECM as a living, dynamic microecosystem (Table 2) [19].

The heterogeneity in the composition of the LME, determined by the diversity of its cellular constituents and acellular compounds, gives rise to distinct protein transcriptomic topography which serves as a critical determinant of the clinical heterogeneity observed among T/NK-cell neoplasms. Moreover, architectural variability within the LME may exert a profound influence on disease prognosis and outcome.

A substantial subset of T/NK-cell neoplasms is associated with EBV infection, affecting either the neoplastic or the stromal compartment. Such infection may induce transcriptomic perturbations in both cellular groups, culminating in altered antigenic expression and reprogrammed cytokine secretion. These processes contribute to dynamic remodeling of the LME, which may consequently modulate the trajectory of lymphomagenesis in either a facilitative or suppressive manner. Analogous mechanisms are implicated in HTLV-1–associated T/NK-cell neoplasms, particularly in ATLL [20].

Given the pronounced micromorphological diversity of T/NK-cell neoplasms, encompassing differences in morphological appearance, growth architecture, neovascularization patterns, immune checkpoint receptor expression, and stromal cell composition, it is plausible to infer that the LME exhibits both structural and functional heterogeneity across disease subtypes. In this context, delineating the precise LME profiles of individual entities represents an increasingly intricate and challenging endeavor. Ultimately, this complexity may underlie the marked variability in clinical behavior and outcome observed not only across distinct T/NK-cell neoplasms but also among nodal, extranodal, and leukemic counterparts of the same nosological category.

In Table 3, we summarize the principal characteristics of T/NK-cell neoplasms, accompanied by an integrative overview of their corresponding LMEs.

## 3. Cellular Stroma Composition in T/NK-Cell Neoplasms

The cellular stroma represents a complex assemblage of non-neoplastic cells intimately admixed with malignant lymphoma cells, collectively establishing the characteristic cytoarchitectural organization of T/NK-cell neoplasms. In conjunction with the ECM, the cellular stroma constitutes the LME, a highly dynamic and functionally diverse compartment.

Based on predominant cellular constituents and biological functions, the LME can be broadly delineated into three principal stromal categories: an angiogenic stroma, composed mainly of endothelial cells and pericytes; a classical stromal compartment, encompassing dendritic cells (DCs), lymphoma-associated macrophages (LAMs), mesenchymal stromal cells (MSCs), myeloid-derived suppressor cells (MDSCs), cancer-/lymphoma-associated fibroblasts (CAFs), and mast cells (MCs); and an immune-cellular stroma, characterized by reactive immune elements including T, B, and NK-cells.

The cellular stromal composition varies significantly among T/NK-cell lymphoma subtypes, highlighting that these differences contribute to distinct clinical characteristics and treatment outcomes for each T/NK-cell neoplasm.

### 3.1. Angiogenic Stroma (Angiogenesis-Neovascularisation)

Angiogenesis is a fundamental process in lymphoma growth and progression. The formation of new blood vessels is driven by pro-angiogenic factors, which are produced by both lymphoma cells and various stromal infiltrating cells, primarily LAMs [21]. In PTCL-NOS, a profile of cytokine expression promotes the polarization of LAMs into M2 macrophages, which stimulate angiogenesis via production of vascular endothelial growth factor (VEGF) and other vasculogenic factors [22]. In fact, in PTCL-NOS, AITL, and ATLL, the majority of the LAMs have been demonstrated to be macrophages with M2 phenotype [23]. T/NK-cell lymphomas with a markedly cellular inflammatory background show more prominent vascular proliferation, which is particularly emphasized in AITL, PTCL, or ENKTCL [24].

The morphologic hallmark of AITL is striking arborized vascularity caused by a CD4^+^ T follicular helper tumor cell clone in orchestration with a robust microenvironment. The proliferation of high endothelial venules (HEV), follicular dendritic cells, and polymorphous reactive cell infiltrate often obscures the minor tumor cell population, rendering the diagnosis quite challenging [25,26]. Increased expression of VEGF type A, a major angiogenic stimulator, was detected in both the tumor cells and endothelial cells, indicating an active role of microvasculature in AITL progression [27].

PTCL-NOS is characterized by histologically unremarkable moderate angiogenesis; however, high VEGF expression has been linked with poor prognosis [28,29]. The major histologic feature of ENKTCL is angiocentric and angiodestructive growth, together with necrosis. The gene expression profiling study in this aggressive lymphoma revealed alterations in angiogenic pathways and overexpression of VEGF-associated genes [30]. A recent study suggested that silencing of IL-33 inhibited ENKTCL angiogenesis by inactivating the Wnt/β-catenin signaling pathway, and identified IL-33 as a prospective therapeutic target [31].

Studies in B-cell NHL indicated an association with increased microvessel density (MVD) and aggressiveness of the disease [32]. In CTCL, microvascular density was significantly higher compared to normal skin with a benign lymphoid infiltrate, which indicated a possible role of angiogenesis in the development of CTCL [33].

The addition of antiangiogenic agents, including bevacizumab and VEGF inhibitors, to conventional therapeutic strategies has yielded conflicting results in several types of T/NK-cell lymphoma [34,35]. A recent study in a high-grade B-cell lymphoma animal model suggested that the activation of angiogenesis in lymphomas differs significantly from that in solid tumors [36]. It was found that non-canonical signaling pathways exerted via VEGFR-3 and LTβ-receptor drive a distinct morphogenic pattern of vasculature formation, while classic, major signaling pathways (HIF-1α or Notch) are not activated. This may have profound implications for anti-angiogenic therapy in lymphoma and certainly warrants further studies.

### 3.2. Stromal Cells

DCs are professional antigen-presenting cells (APCs) that bridge innate and adaptive immunity by recognizing tumor or viral antigens and activating naïve CD8^+^ and CD4^+^ T-cells [37]. In ENKTCL, DCs are exposed to tumor, and EBV-derived immunosuppressive factors, acquiring a tolerogenic phenotype that impairs antigen presentation and weakens T-cell priming, enabling immune evasion and tumor progression [38]. The absence of immune pressure accelerates tumor proliferation and the accumulation of additional oncogenic alterations.

Within the ENKTCL microenvironment, cytokines such as IL-10, TGF-β, and PGE2 inhibit DC maturation, maintaining them in an immature state that promotes immune tolerance rather than activation [39,40,41]. Tolerogenic DCs secrete IL-10, TGF-β, and indoleamine 2,3-dioxygenase (IDO), leading to tryptophan depletion, T-cell arrest, Treg expansion, and apoptosis, creating a profoundly immunosuppressive niche [42,43,44]. Tumor-derived and EBV-associated factors (e.g., LMP1) further downregulate MHC I/II and co-stimulatory molecules CD80/CD86, preventing effective T-cell priming [37,45].

Functionally, ENKTCL-associated DCs fail to secrete cytotoxic lymphocyte-recruiting chemokines (CXCL9, CXCL10, CCL5) and may instead attract Tregs or MDSCs, resulting in reduced CD8^+^ T-cell and NK-cell infiltration, an “immune-cold” phenotype resistant to checkpoint blockade [46,47,48].

Therapeutically, reversing DC dysfunction through vaccines, TLR agonists, FLT3 ligands, or combination with immune checkpoint inhibitors may enhance antitumor immunity and improve treatment efficacy in ENKTCL.

LAMs are myeloid-lineage cells derived from circulating monocytes, which are recruited into the LME by tumor, and EBV-derived factors such as CCL2, colony-stimulating factor 1 (CSF-1), and inflammatory cytokines [21,49]. Within the LME, LAMs predominantly adopt an M2-like (alternatively activated) phenotype, expressing markers such as CD163, CD206, and arginase-1, and producing anti-inflammatory cytokines including IL-10 and TGF-β [50,51,52]. This polarization is further reinforced by hypoxia, metabolic stress, and signals from malignant T/NK-cells [50]. M2-type LAMs promote angiogenesis, lymphangiogenesis, tissue remodeling, and tumor cell proliferation, while suppressing adaptive immune responses, contributing to drug resistance, aggressive histology, and poor clinical outcomes [21,53,54].

A key immunosuppressive mechanism of LAMs involves the induction and maintenance of regulatory T-cells (Tregs). IL-10 inhibits effector T-cell activation and proliferation and promotes the differentiation of naïve CD4^+^ T-cells into inducible Tregs, while TGF-β drives FOXP3 expression and stabilizes Treg suppressive function. LAMs also express indoleamine 2,3-dioxygenase (IDO), which depletes tryptophan and generates kynurenine metabolites, arresting T-cell proliferation, inducing apoptosis, and further promoting Treg expansion. In addition, chemokines such as CCL17 and CCL22 selectively recruit Tregs via CCR4, creating a profoundly immunosuppressive niche that inhibits CD8^+^ cytotoxic T-cells and NK-cells and facilitates immune escape [55].

In EBV-driven neoplasms such as ENKTCL, LAMs not only secrete immunosuppressive cytokines but also express PD-L1, induced by EBV-mediated oncogenic signaling (e.g., LMP1) and inflammatory mediators such as IL-10 and IFN-γ [56]. PD-L1^+^ macrophages reinforce T-cell dysfunction and contribute to resistance to PD-1/PD-L1 checkpoint blockade [57]. Beyond immunosuppression, LAMs promote tumor growth through secretion of vascular endothelial growth factor (VEGF) and epidermal growth factor (EGF), supporting angiogenesis, extracellular matrix remodeling, and recruitment of additional stromal elements [53,54].

Clinically, high densities of CD68^+^ or CD163^+^ LAMs in the LME are strongly associated with poor progression-free survival (PFS) and overall survival (OS) in patients with mature T/NK-cell lymphomas [58,59]. T/NK-cell neoplasms are typically characterized by hyperinflammatory stroma with hyperactive macrophages. The characteristic cytokine spectrum of M2-type macrophages is high expression of IL-10 and low expression of IL-12. Increased LAM infiltration correlates with B symptoms, extranodal involvement, chemoresistance, and impaired response to immunotherapy [60]. In PTCL-NOS, malignant T-cells drive LAM proliferation via the JAK/CSF1R signaling axis, and experimental dual inhibition of these pathways reduces macrophage expansion, decreases disease burden, and prolongs survival in preclinical models [61].

Taken together, LAMs represent central regulators of the LME in T/NK-cell lymphomas, integrating tumor-promoting, immunosuppressive, and stromal-supportive functions. Therapeutic strategies targeting LAM-mediated mechanisms, including cytokine blockade (IL-10, TGF-β), metabolic inhibition (IDO), PD-L1 blockade, and disruption of JAK/CSF1R-driven proliferation offer promising avenues to restore antitumor immunity, enhance the efficacy of immunotherapies, and improve clinical outcomes in these aggressive malignancies.

The MSCs are multipotent cells found in various tissues, including bone marrow, adipose tissue, and the stroma of lymphoid organs and could exhibit both anti-inflammatory, and immunosuppressive properties [49,62]. MSCs within the ENKTCL microenvironment can inhibit anti-tumor immune responses through multiple mechanisms. They secrete immunosuppressive cytokines such as IL-10, TGF-β and PGE2, which impair DCs maturation, reduce cytotoxic T lymphocyte and NK-cell function, and promote Treg expansion [63]. These factors create an immunosuppressive milieu that enables EBV-infected NK/T-cells to evade immune surveillance. Furthermore, MSCs may express or induce the expression of immune checkpoint ligands, including PD-L1, on themselves or on other immune and stromal cells, thereby contributing to T-cell exhaustion, and resistance to immune checkpoint inhibitors [63].

ENKTCL-derived or tumor-educated MSCs can produce a range of growth factors and extracellular matrix components that facilitate tumor cell proliferation, invasion and survival, such as VEGF, hepatocyte growth factor (HGF) and stromal-derived factor 1 (SDF-1/CXCL12). The interaction between MSCs and malignant lymphoma cells may also involve exosome-mediated communication, in which MSC-derived exosomes carry microRNAs, cytokines, and other molecules that alter gene expression and promote tumor progression [64]. MSCs have been implicated in the development of chemoresistance in various hematological malignancies, including lymphoma. In ENKTCL, MSCs may protect tumor cells from apoptosis by secreting soluble anti-apoptotic factors (e.g., IL-6, CXCL12), altering drug metabolism in the LME and physically shielding tumor cells through the formation of protective stromal niches [65,66]. These mechanisms reduce the efficacy of cytotoxic agents and contribute to the refractory nature of ENKTCL.

Tumor cells and the inflammatory microenvironment can actively recruit MSCs from bone marrow or surrounding tissues. This recruitment is mediated by chemokines such as CCL2, CXCL8 and CXCL12, as well as factors secreted by EBV-infected cells [67]. Once in the LME, MSCs may differentiate into CAFs or other supportive stromal phenotypes that further enhance tumor growth, and immune escape [68].

In ENKTCL, mesenchymal stromal cells are key contributors to the establishment and maintenance of an immunosuppressive and tumor-supportive microenvironment. Through cytokine secretion, immune modulation, metabolic support, and physical interactions with tumor cells, MSCs help drive disease progression, immune evasion, and treatment resistance [69]. Targeting MSC-tumor interactions or reprogramming stromal cell functions may offer new therapeutic avenues to improve outcomes in this aggressive lymphoma.

MDSCs are a heterogeneous population of immature myeloid cells that expand under pathological conditions such as cancer, chronic infection, and inflammation, and possess potent immunosuppressive activity [70]. In ENKTCL, MDSCs are recruited and activated within the tumor microenvironment (TME) by EBV- and tumor-driven factors, including GM-CSF, IL-6, TGF-β, CCL2, and CXCL8, which promote both monocytic (M-MDSC) and granulocytic (PMN-MDSC) subtypes and enhance their suppressive phenotype [71,72,73].

MDSCs inhibit antitumor immunity through multiple mechanisms. Arginase-1 (ARG1) depletes extracellular L-arginine, downregulating the CD3ζ chain of T-cell receptors and arresting T-cell proliferation [74,75,76]. Inducible nitric oxide synthase (iNOS) produces nitric oxide, disrupting TCR signaling and inducing T-cell apoptosis [77]. IDO degrades tryptophan into kynurenine, suppressing T-cell responses and promoting Treg differentiation [72,78]. Additionally, reactive oxygen species (ROS) generated by MDSCs induce oxidative stress in CD8^+^ T-cells and NK-cells, impairing cytotoxic function [79]. MDSC-derived IL-10 and TGF-β further suppress dendritic cell maturation, reduce pro-inflammatory cytokine production, and expand FOXP3^+^ Tregs, reinforcing immunosuppressive networks within the LME [80,81].

CAFs, although primarily studied in solid tumors and B-cell lymphomas, are increasingly recognized in T/NK-cell neoplasms as modulators of the microenvironment. CAFs remodel the ECM, secrete IL-6, CXCL12, and TGF-β, and cooperate with malignant cells to establish pro-inflammatory and immunosuppressive niches, intersecting with key pathways such as JAK/STAT and NF-κB and enhancing immune evasion via PD-L1 upregulation [82,83,84,85].

Although direct functional data for T/NK-cell neoplasms are limited, the EBV-driven cytokine milieu, characterized by IL-6 and IL-10, likely amplifies CAF-like mechanisms, linking stromal remodeling, inflammation, and immune suppression. WHO-HAEM5 formally recognizes stroma-derived neoplasms, underscoring the clinical and biological relevance of fibroblastic elements in lymphoid malignancies [4]. Collectively, MDSCs and CAFs represent central stromal hubs where immune escape, inflammation, and tissue remodeling converge, highlighting potential therapeutic targets in T/NK-cell lymphomas [82].

MCs are clearly significant in certain T/NK-cell lymphomas (especially AITL, CTCL) where they help drive inflammation, angiogenesis, Th17 cell presence, and worse disease behavior. For NK/T-cell lymphomas specifically, there is little direct evidence so far for mast cell involvement, so this is an open area. MCs actively contribute to angiogenesis and induce neovascularization by releasing the classical proangiogenic factors including VEGF, FGF-2, PDGF, and IL-6, and nonclassical proangiogenic factors mainly proteases including tryptase and chymase. MCs support tumor invasiveness by releasing a broad range of matrix MMPs [86].

### 3.3. Immuno-Cellular Stroma (Reactive T, NK and B Lymphocytes)

The LME of T/NK-cell lymphomas comprises reactive immune cells: CD8^+^ cytotoxic T-cells, Tregs, reactive NK-cells, T_FH_, and CD4^+^ helper T-cells, forming the immune-cellular stroma. Malignant tissue mainly consists of neoplastic T/NK-cells, while data on reactive counterparts remain scarce. Immune subtyping divides these neoplasms into “inflamed” and “immune-suppressed” categories, with T-cell exhaustion linked to poor outcomes [87,88].

Reactive CD8^+^ T-cells exhibit antitumor activity mainly in early disease, whereas Th2 polarization suppresses cytotoxicity via IL-4, IL-5, and IL-13, and reduced IFN-γ/IL-12 [89]. High CD8^+^ T-cell counts correlate with better prognosis in some CTCLs.

Tregs foster immunosuppression by secreting IL-10, IL-35, TGF-β, expressing PD-1, CTLA-4, LAG-3, and depleting IL-2 [90]. They suppress multiple immune subsets. Wang et al. identified four Treg subsets, suppressor, tumor-killing, malignant, and incompetent, with their proportions influencing prognosis; e.g., in AITL, incompetent or tumor-killing Tregs correlate with better outcomes [91].

T_FH_ cells, expressing CXCL13, PD-1, ICOS, and BCL6, are diagnostic for AITL and related PTCL-NOS variants [92]. The roles of T_FH_ and CD4^+^ helper cells remain unclear.

Reactive NK-cells, difficult to distinguish from malignant counterparts, can be evaluated by KIR repertoire diversity (polyclonal = reactive) [93,94]. In T/NK neoplasms, reactive NK-cells often exhibit functional exhaustion, sometimes EBV-infected, losing cytotoxic capacity [93,94,95].

Among T/NK-cell neoplasms, ENKTCL is best characterized. It displays high stromal heterogeneity shaped by EBV, which enhances PD-L1 expression and alters cytokine/chemokine signaling [57]. T and NK-cells show exhaustion (PD-1, TIM-3, LAG-3 expression) and are functionally depleted. Malignant NK-cells secrete dipeptidyl peptidase-4 (DPP4), suppressing chemotaxis of normal NK/T-cells and eosinophils; DPP4 inhibition may restore Th1 activity and synergize with PD-1 blockade [96].

Antigen presentation is impaired by defective DC maturation and PD-L1–high DCs, promoting T-cell anergy [97]. JAK3/STAT3 mutations further induce PD-L1, reinforcing immune evasion [38,88]. Immune checkpoint inhibitors show durable but limited efficacy in ENKTCL [98,99,100]; combining them with DPP4 or JAK inhibitors or adoptive T-cell therapy may improve outcomes.

Bystander B-cells represent non-malignant (reactive) B lymphocytes residing within the LME of T/NK-cell lymphomas. These cells are frequently infected by EBV, facilitating complex interactions with neoplastic lymphoma cells and other immune constituents. Through these interactions, bystander B-cells may modulate the biological behavior of the malignancy, influencing both disease progression and therapeutic responsiveness, although their precise functional role remains to be fully elucidated [101]. It has been demonstrated that reactive B-cells secrete anti-inflammatory cytokines, including IL-10 and TGF-β, thereby fostering a pro-tumorigenic microenvironment via suppression of T-cell–mediated immune responses [102].

Among T-cell lymphomas, AITL constitutes a prototypic example characterized by a distinctive microarchitectural organization enriched with scattered EBV^+^ bystander B-cells within a reactive background. This histopathological feature is associated with a propensity toward secondary aggressive B-cell lymphomagenesis, particularly diffuse large B-cell lymphoma (DLBCL), during the clinical course of long-standing AITL [103]. Analogous observations have been reported in peripheral T-cell lymphoma, not otherwise specified (PTCL-NOS), in which the presence of EBV^+^ bystander B-cells correlates with similar disease evolution culminating in DLBCL transformation [104].

In the context of CTCL, infiltration by reactive B-cells appears to exert a biologically relevant impact, demonstrating significant associations with advanced disease stages and reduced PFS [105]. In contrast, within ENKTCL, a high density of bystander B-cell infiltration has been paradoxically linked to favorable clinical outcomes, including improved OS, observed in approximately 40% of affected patients [106].

## 4. Acellular Stroma-Extracellular Matrix Composition in T/NK-Cell Neoplasms

The acellular stroma comprises amorphous substances and protein fibers secreted by stromal (accessory) cells in concert with lymphoma cells, together forming the ECM. Figure 1 has illustrated the composition of the ECM in earlier section. In this context, the terms acellular stroma and ECM are largely synonymous. The amorphous component consists primarily of glycoproteins and proteoglycan complexes, while collagen and elastin fibers provide structural support, creating a three-dimensional scaffold. Incorporation of functionally active biomolecules—including remodeling enzymes and their inhibitors, cytokines, hormones, exosomes, and other mediators—completes the fully functional ECM architecture.

The ECM represents a complex, dynamic, and interactive medium in which stromal and lymphoma cells form an inseparable ensemble essential for tissue homeostasis. It provides structural support and regulates cell behavior through biochemical and mechanical cues, mainly via interactions with integrins [107]. Acting as a reservoir for bioactive molecules, the ECM releases growth factors and cytokines upon disruption, influencing the local microenvironment [108].

Tissue development depends on the ECM’s control of proliferation, migration, differentiation, and apoptosis [107]. Its major structural proteins, collagens and elastin, ensure tensile strength and elasticity. Collagen, comprising about 30% of total body protein, includes at least 28 subtypes, while elastin maintains tissue extensibility [109].

Glycoproteins—such as fibronectin, laminin, tenascin, and thrombospondin—mediate adhesion, migration, and signaling between cells and the ECM. Fibronectin provides scaffolding for ECM assembly; laminin stabilizes basement membranes; and tenascins and thrombospondin modulate adhesion and cell–matrix interactions [110,111].

Proteoglycans consist of core proteins linked to glycosaminoglycan (GAG) chains that confer compressive resistance and interact with growth factors. Large proteoglycans include aggrecan and versican, while small leucine-rich proteoglycans (decorin, biglycan, fibromodulin, lumican) regulate tissue integrity and ECM organization.

Matrix metalloproteinases (MMPs) are zinc-dependent endopeptidases that remodel the ECM in physiological and pathological contexts [112,113]. They release growth and angiogenic factors, cleave receptors, and modulate cytokines [114,115]. Their activity is controlled by tissue inhibitors of metalloproteinases (TIMPs), which also have MMP-independent roles in proliferation and apoptosis. Dysregulation of the MMP–TIMP balance may drive excessive ECM degradation, offering potential therapeutic targets [109].

Altogether, ECM components critically shape the LME and influence the lymphomagenesis of T/NK-cell neoplasms, emphasizing the need to clarify their individual and collective functions.

### 4.1. Role of Extracellular Matrix Protein Fibers in T/NK-Cell Lymphomagenesis

Data on the precise role of collagens and other ECM fibers in the pathogenesis of T/NK-cell neoplasms remain limited. Most available evidence derives from studies across diverse malignancies rather than T/NK-cell lymphomas specifically. In general, tumor-specific ECM is characterized by increased collagen density, resulting in greater tissue stiffness, which has been correlated with poor prognosis in multiple cancer types, although the mechanistic basis for this association remains incompletely understood. Notably, enhanced collagen fibrosis is infrequently observed in T/NK-cell neoplasms, with the possible exception of ENKTCL.

Studies suggest that collagen may facilitate cancer cell proliferation and migration while simultaneously modulating the function and phenotype of tumor-infiltrating immune cells, including lymphoma-associated macrophages (LAMs) and T-cells. This indicates that tumor-associated collagen may exert immune-regulatory effects within the tumor microenvironment, influencing both disease progression and responsiveness to immunotherapy. The ability of the ECM to shape immune cell behavior has given rise to the emerging field of matrix immunology.

Current evidence largely focuses on type I collagen, the most abundant collagen isoform, though other collagen types may exert distinct effects on immune cells [116]. Both transmembrane and extracellular collagens produced by tumor and stromal cells can engage leukocyte-associated immunoglobulin-like receptor-1 (LAIR-1, CD305), activating inhibitory signaling pathways. Overexpression of collagen type XVII on target cells has been shown to reduce NK-cell cytotoxicity. Collectively, these findings suggest that tumor-expressed collagens contribute to immune evasion by directly modulating T and NK-cell activity or indirectly affecting other microenvironmental components such as macrophages [117].

Excessive collagen deposition contributes to ECM stiffness, which may enhance tissue fibrosis and impair drug delivery to lymphoma cells [118]. In ENKTCL, fibrosis can promote necrosis and angio-destructive processes, occasionally leading to pseudo-epithelial hyperplasia—a characteristic histopathologic feature—and may contribute to treatment resistance [119,120]. Collagen can also impede NK-cell-mediated cytotoxicity by promoting a shift toward cytokine production and by facilitating the formation of a protective glycocalyx on cancer cells, further enabling immune evasion. Moreover, cancer cells can remodel collagen to reinforce tumor progression, enhancing adhesion and migration. Consequently, targeting collagen deposition in the tumor microenvironment may augment NK-cell cytotoxicity and represent a potential therapeutic strategy. Clinically, the density and composition of collagen within tumors often correlate with prognosis, with higher collagen levels generally indicating poorer outcomes.

The role of elastin fibers in T/NK-cell lymphomagenesis has not been systematically investigated. Available data pertain primarily to ENKTCL, where lymphoma cells exhibit angiocentric growth patterns that disrupt and destroy elastic fibers within blood vessel walls. This process, accompanied by fibrinoid deposition and ischemia, contributes to the characteristic necrosis observed histologically. Widespread infiltration of lymphoma cells into the elastic lamina of small arteries further defines the distinctive histopathologic landscape of ENKTCL [121].

### 4.2. Role of Extracellular Matrix Glycoproteins in T/NK-Cell Lymphomagenesis

Glycoproteins are among the most dysregulated components of the cancer matrisome. Certain glycoproteins are consistently up or downregulated across multiple malignancies compared with normal tissues, suggesting their involvement in general mechanisms of tumor progression [19].

Fibronectin, for instance, exhibits altered expression in T/NK-cell neoplasms as well as in other cancers, promoting tumor growth, migration, invasion, and resistance to therapy [122]. Within the lymphoid system, fibronectin is typically absent in healthy lymph nodes but is overexpressed in neovasculature associated with tumors, including lymphomas. Notably, the extra domain B (ED-B) isoform is frequently localized to the lymphoma-associated subendothelial ECM and serves as a recognized angiogenic marker. Overexpression of ED-B fibronectin has been documented across various lymphoma types, representing a potential therapeutic target. In a xenograft model of B-cell NHL, Schliemann et al. demonstrated that a fusion protein, L19-IL2, in combination with rituximab, could achieve complete eradication of lymphoma cells [123]. Accordingly, ED-B fibronectin constitutes a promising target for monoclonal antibody-based therapies aimed at both visualization and treatment of lymphomas. In T/NK-cell neoplasms, fibronectin contributes to tumor aggressiveness and survival by modulating the microenvironment, facilitating cell adhesion, migration, and resistance to therapy [122]. While direct targeting of fibronectin is not yet standard practice, disruption of fibronectin–integrin interactions may represent a viable translational approach to compromise the survival niche of T/NK lymphoma cells.

Data on laminin involvement in T/NK-cell lymphomagenesis are limited. Gene expression analyses (e.g., LAMB2, LAMC2) suggest potential roles in the T-cell lymphoma microenvironment, and ECM studies confirm its presence, albeit to a lesser extent relative to other matrix proteins. Laminin isoforms are established modulators of T-cell behavior within lymphoid tissues and may indirectly influence lymphoma biology. Some evidence indicates laminin can inhibit NK-cell-mediated tumor cytotoxicity through interactions with both NK cells and tumor cells, while other studies suggest NK cells can produce laminin, contributing to regulation of tumor invasion [124]. The oncofetal antigen immature laminin receptor protein (OFA-iLRP) is highly conserved, expressed in fetal tissues and various cancers, including hematopoietic malignancies, but absent in physiologically differentiated adult cells. OFA-iLRP represents a potential target for T-cell-based immunotherapy in hematologic malignancies [125]. Similarly, the laminin receptor (LR) is overexpressed in neoplastic cells relative to normal counterparts and may serve as a biomarker of metastatic aggressiveness across multiple cancers, including leukemia and lymphoma [126].

Tenascin-C is overexpressed in both embryonic and adult ECM, particularly in tumor tissues [127,128]. Its expression correlates spatially and temporally with tumor neovascularization and may confer anti-adhesive and immunosuppressive properties, thereby promoting lymphoma cell survival, migration, and angiogenesis [129,130,131,132]. In a study using the monoclonal antibody tenatumomab across 100 patients with T/NK-cell neoplasms (75 PTCL; 25 CTCL), tenascin-C expression was observed in multiple subtypes, including ALCL, ALK-negative (*n* = 21), ALCL, ALK-positive (n = 19), PTCL-NOS (n = 20), MF (n = 13), AITL (n = 9), CD30^+^ primary CTCL (n = 6), and other subtypes (n = 12). While expression intensity varied, no statistically significant differences were observed (*p* = 0.334). A high proportion of tenascin-C expression (>50%) was noted in ALCL, ALK^−^ (81%), AITL (78%), and ALCL, ALK^+^ (58%), whereas PTCL-NOS (30%) and CTCL (24%) showed lower expression (*p* = 0.0019). Gene expression datasets confirmed significant tenascin-C overexpression in T/NK-cell neoplasms compared to normal tissues [128]. Histologic subtypes such as ALCL and AITL exhibited strong, diffuse tenascin-C staining, whereas MF/SSy and primary cutaneous ALCL displayed less intense, sparse staining. These differences likely reflect tissue remodeling and neoangiogenesis characteristic of aggressive PTCL. Notably, vascular-associated tenascin-C expression may influence survival outcomes, highlighting its potential as a therapeutic target, as evidenced by the activity of radiolabeled tenatumomab [133,134].

### 4.3. Role of Proteoglycans in T/NK-Cell Lymphomagenesis

Proteoglycans are key effectors within the pericellular zone in both healthy and malignant tissues. In lymphoid malignancies, increased mRNA and protein expression of serglycin has been detected in malignant cells, suggesting that proteoglycan synthesis accompanies lymphoid transformation [135]. Syndecan-4 is overexpressed in malignant T cells from patients with SSy, implicating it in disease pathogenesis and representing a potential therapeutic target [136]. Similarly, aberrant upregulation of versican isoform V1 by Sézary cells has been associated with enhanced migration and cutaneous tropism, while potentially sensitizing cells to chemotherapeutics [137,138].

Proteoglycans exhibit context-dependent effects in cancer progression and metastasis. Certain proteoglycans, such as syndecan-1 and glypican-1, promote tumor growth and dissemination by enhancing growth factor signaling and cell migration, whereas others, including decorin and lumican, inhibit tumor progression by modulating immune responses and collagen fibrillogenesis. Collectively, proteoglycans contribute to the regulation of microenvironmental inflammation and immunity, influencing both lymphoma biology and therapeutic responses [139].

### 4.4. Role of MMPs and TIMPs in T/NK-Cell Lymphomagenesis

Dysregulated activity of matrix metalloproteinases (MMPs) has been recognized as a key contributor to cancer dissemination by degrading the ECM components and facilitating cellular migration. Both MMPs and their endogenous inhibitors, tissue inhibitors of metalloproteinases (TIMPs), play critical roles in the development and progression of various lymphomas. Specifically, MMP-2 and MMP-9 are central to ECM remodeling across multiple lymphoma subtypes. Elevated expression and activation of these enzymes are associated with tumor invasion, metastasis, and poor clinical outcomes. Mechanistically, MMP overexpression may be induced by interactions with endothelial cells, accelerating lymphoma progression.

Among T/NK-cell neoplasms, evidence indicates that MMP dysregulation significantly impacts disease behavior in select entities. MMP-9 overexpression is particularly characteristic of ENKTCL, where it degrades multiple ECM substrates—including fibronectin, laminin, collagen, elastin, and casein—contributing to the tumor’s high propensity for dissemination and the extensive necrosis commonly observed [140,141]. ENKTCL is typified by vascular invasion and ECM destruction; elevated MMP-9 may exacerbate tissue necrosis via enhanced angiodestruction and may contribute to chemoresistance by impairing drug delivery [140,142]. Additional proteolytic activity by MMP-1 and MMP-11 may be linked to EBV-driven mechanisms. High expression levels of MMP-26 and MMP-9 have also been associated with disease invasiveness and progression, and these enzymes may serve as biomarkers to distinguish ENKTCL from reactive lymphoid hyperplasia, highlighting their potential prognostic value [143].

In contrast, expression of MMP-2 and MMP-9 in other lymphoma subtypes appears less pronounced than in epithelial malignancies, likely reflecting differences in stromal composition; fibroblasts dominate ECM remodeling in carcinomas, whereas lymphoid stroma is more cellularly heterogeneous. Synthetic agents that modulate TIMP activity or inhibit cytotoxic granule secretion have been proposed as therapeutic alternatives, though the dual and context-dependent functions of TIMPs have limited their clinical utility to date [144].

Dysregulated ECM contributes to lymphoma progression through both direct and indirect mechanisms. Directly, ECM alterations influence malignant cell transformation, expansion of cancer stem cells, and disruption of tissue polarity, facilitating invasion and metastasis. Indirectly, ECM remodeling affects stromal cells, promotes angiogenesis and inflammation, and establishes a tumor-permissive microenvironment [145,146,147]. In lymphomas, ECM dysregulation is characterized by vascular disorganization, enlarged vessel pores driven by VEGF, PDGF-β, and TGF-β overexposure, hypoxia, and impaired systemic immune cell infiltration. Locally, immune cells secrete pro-proliferative cytokines: M2-polarized macrophages produce IL-10, T_FH_ cells secrete IL-21, and Th17 and CD8^+^ T-cells produce IL-6, collectively supporting lymphoma cell survival and invasion.

Despite the limited success of early-phase clinical trials with broad-spectrum MMP inhibitors, dysregulated MMP activity—particularly in early disease stages—remains an area of active investigation. Future therapeutic strategies are likely to focus on the selective inhibition of individual MMP family members to mitigate off-target effects and improve clinical outcomes [148].

### 4.5. Role of Cytokines, Cytokine Receptors, Growth Factors in T/NK-Cell Lymphomagenesis

Cytokines constitute pivotal intercellular mediators that orchestrate communication among the diverse cellular constituents of lymphomas. Their expression patterns are highly heterogeneous and frequently correspond to specific T/NK-cell neoplasm subtypes. Intriguingly, such heterogeneity may also manifest within tumors of identical histological classification. Functionally, cytokines critically influence the inflammatory microenvironment, a topic that will be elaborated in the dedicated section on inflammation. Among these, most notably IL-6 play a central role in establishing and maintaining an inflammatory milieu that supports lymphoma cell survival and progression.

### 4.6. Role of Exosome or Lymphoma Cell Extracellular Vesicles in T/NK-Cell Lymphomagenesis

Lymphoma cell–derived extracellular vesicles (LCEVs) have been assigned a multifaceted role in the regulation of lymphoma homeostasis. Structurally, similar to other exosomes, LCEVs possess a bilayer lipid membrane enriched with surface-targeting molecules such as tetraspanins (CD9, CD63, and CD81), major histocompatibility complex (MHC) class I and II proteins, and integrins. Their internal cargo includes various nucleic acids, DNA harboring fusion genes (e.g., EML4-ALK), long non-coding RNAs (lncRNAs), circular RNAs (circRNAs), and microRNAs (miRNAs), as well as proteins such as soluble NSF attachment protein receptor (SNARE), annexin, flotillin, ALG-2–interacting protein X (Alix), tumor susceptibility gene 101 (TSG101), and mutated variants including mutMYD88 [149,150,151].

Historically, exosomes were regarded as cellular waste products released into the ECM. However, current evidence attributes to LCEVs a broad modulatory function within the LME, where they influence immune evasion, therapeutic response, and drug resistance, and may also serve as novel “multi-omic vesicles” for disease detection [152]. Increasing attention has been directed toward LCEV-mediated interactions between malignant cells, immune cells, and other LME constituents as pivotal drivers of lymphoma progression and treatment outcomes [153].

In EBV^+^ NK-cell lymphoproliferative disorders (LPDs), it has been proposed that EBV^+^ memory B-cells continuously secrete exosomes that potentiate the immunosuppressive effects of infected cells, promote clonal proliferation of EBV^+^ T/NK-cells to variable extents, and thereby contribute to the heterogeneous clinical course and prognosis of these entities [154]. Cumulative evidence indicates that lymphoma-derived exosomes critically contribute to lymphomagenesis, disease progression, and treatment resistance [155,156,157]. Among the best-characterized immune evasion pathways, the programmed cell death protein 1/programmed death ligand 1 (PD-1/PD-L1) axis plays a central role [158]. Elevated levels of circulating and membrane-bound PD-L1 have been identified as adverse prognostic indicators and potential diagnostic and prognostic biomarkers in ENKTCL [159].

In summary, the extensive release of exosomes by lymphoma and immune cells represents a key mechanism in sustaining lymphoma progression and shaping the immunoregulatory landscape of the LME.

## 5. Role of the Biological Agencies (EBV and HTLV-1) in T/NK-Cell Lymphomagenesis

It has long been established that EBV primarily infects B lymphocytes as well as various stromal components surrounding neoplastic T or NK-cells. More recent evidence, however, has demonstrated that EBV is also capable of directly infecting T and NK-cells themselves. This phenomenon contributes to the molecular reprogramming of these cells within the context of the surrounding LME, thereby influencing lymphoma pathobiology. In parallel, HTLV-1 exhibits a distinct CD4^+^ T-cell tropism, leading to direct infection of these lymphocytes. The resultant molecular alterations closely parallel those observed in EBV-mediated transformation, highlighting convergent mechanisms of virus-driven lymphomagenesis.

### 5.1. EBV as the T/NK-Cell Lymphoma Promoting Agent

EBV is nearly ubiquitous, with serologic evidence of prior infection detected in more than 90% of adults worldwide. Despite this widespread exposure, the incidence of T/NK-cell lymphomas remains low, indicating that EBV infection alone is insufficient to induce malignant transformation. Rather, EBV likely acts as an initiating event that increases susceptibility to oncogenesis in a permissive host environment.

The significantly higher prevalence of T/NK-cell malignancies in EBV-endemic regions further supports the contribution of additional cofactors. The interplay between extrinsic (environmental) and intrinsic (genetic) determinants appears critical in driving tumorigenesis, reinforcing the concept of cancer as a multifactorial disease process.

Histologic and molecular analyses of paraffin-embedded tumor specimens have strengthened the evidence for a viral role in oncogenesis. Detection of EBER transcripts by ISH remains a diagnostic hallmark for identifying EBV-associated T/NK-cell neoplasms. Cumulative molecular data now support EBV as an early oncogenic driver in a subset of nodal, extranodal, and leukemic T/NK-cell entities. Moreover, EBV genome sequencing has revealed two major viral strains and several variants of the LMP1, each exhibiting distinct oncogenic potentials that may contribute to the biological and clinical heterogeneity of these lymphomas [2].

EBV displays a strong tropism for B-cells and epithelial cells, with the complement receptor 2 (CD21) serving as the principal entry receptor. However, the mechanism by which T-cells and NK-cells become infected has long remained unclear. One proposed explanation is that EBV infects common lymphoid progenitor cells expressing CD21, which subsequently differentiate into NK-cell and T-cell lineages. This hypothesis provides a plausible basis for the presence of EBV within non–B-cell compartments and may account for the clonal EBV genomes observed in EBV-associated T/NK-cell lymphomas. Nevertheless, recent findings have demonstrated that the type 2 EBV strain possesses a unique tropism enabling infection of mature T-cells, a process that critically depends on the interaction between viral glycoprotein gp350 and the cellular receptor CD21 [160]. Following infection, EBV-transformed T or NK-cells undergo proliferation supported by viral oncoproteins, most notably LMP1, which functions as a constitutive mimic of CD40. Through this mechanism, LMP1 persistently activates multiple signaling cascades, including the AKT, MAPK, JNK, STAT, and NF-κB pathways, thereby enhancing cell-cycle progression, inhibiting apoptosis, and modulating immune responses [161].

These EBV-infected cellular clusters also exhibit elevated expression of immune checkpoint molecules, including PD-L1 and CD86, which mediate profound immunosuppressive interactions with T cells via PD-L1/PD-1 and CD86/CTLA4 signaling axes. This observation suggests that LMP1 may represent a central driver of oncogenic transformation and tumor maintenance within EBV-associated T/NK-cell neoplasms. Furthermore, genomic instability induced by EBV infection contributes to the accumulation of somatic mutations in oncogenes and tumor suppressor genes, thereby promoting the emergence and evolution of EBV-driven T/NK-cell lymphomas [162].

Single-cell transcriptomic analyses have recently delineated at least three major LME phenotypic clusters with distinct immune compositions [163]. The LME1 cluster exhibits an immune-desert phenotype characterized by the absence of T and myeloid dendritic cells (MDCs). The LME2 cluster corresponds to an immune-deficient phenotype, displaying the presence of T and stromal cells but lacking MDCs, indicative of impaired innate immunity. In contrast, the LME3 cluster represents an immune-inflamed phenotype, distinguished by increased infiltration of both T cells and MDCs and associated with signatures of immune activation and exhaustion [163].

Across all three phenotypic clusters, dysregulation of G protein–coupled receptor (GPCR) signaling pathways has been observed, a process modulated by EBV that contributes to the remodeling of cancer immunity and promotes lymphoma progression.

Notably, overexpression of chemokine receptor 1 (CCR1) has been identified in LME1 and LME3, where it contributes to the modulation of immunosuppressive cell populations within the virus–cancer interface of the LME. These findings highlight CCR1 and related pathways as potential targets for therapeutic intervention.

EBV-infected lymphoma cells, together with surrounding stromal components, cooperatively remodel the ECM. This reorganization transforms the ECM into a permissive scaffold that fosters lymphoma cell survival, proliferation, and progression, thereby reinforcing the complex interplay between viral oncogenesis and the tumor microenvironment.

### 5.2. HTLV-1 as the T/NK-Cell Lymphoma Promoting Agent

In contrast to EBV, HTLV-1 is less ubiquitous and exhibits a geographically restricted endemic distribution, with high-prevalence clusters (>5% of tested individuals) in southwestern Japan, West Africa, Central and South America (notably Brazil), and the Caribbean [164]. Surrounding these clusters are intermediate (<5%) and low (~1%) prevalence regions, including the USA, Canada, Australia, Chile, Argentina, India, Iran, and several European countries [165]. These high-prevalence regions demonstrate unusually elevated incidence of ATLL across all four disease subtypes, reflecting a strong causal relationship between HTLV-1 infection and oncogenesis.

HTLV-1 encodes the viral transcriptional transactivator Tax in the pX region of its genome, which plays a central role in malignant transformation. Tax interacts with host cell proteins, modulates intracellular signaling pathways, regulates gene transcription, and drives proliferation of HTLV-1-infected T-cells [166,167]. In asymptomatic carriers, a balance exists between proliferation of infected T-cells and immune clearance by cytotoxic CD8^+^ T-cells. Expression of viral proteins renders HTLV-1-infected cells antigenic, provoking host immunity and necessitating multiple immune escape mechanisms during ATLL development [168,169]. Tax contributes to this immune evasion by modulating immune-related pathways that enhance survival.

Genetic alterations affecting the MHC class I complex (HLA-A, HLA-B, and β2-microglobulin) have been reported in 54% of ATLL cases, indicating that loss of MHC class I–mediated immune recognition is a critical step in pathogenesis [170]. ATLL cells retaining both MHC class I and β2-microglobulin exhibit better clinical outcomes compared to cases lacking these molecules [171]. MHC class II expression, regulated by the class II transactivator, is higher in indolent ATLL than in aggressive subtypes, suggesting a role in disease progression and serving as an independent favorable prognostic marker [172,173,174]. Interaction between MHC class II on tumor cells and CD4^+^ tumor-infiltrating lymphocytes (TILs) may further contribute to immune evasion [168].

Profound immunodeficiency is characteristic of ATLL, facilitating the accumulation of genetic abnormalities in immune-related genes and selection of clones capable of escaping host immunity. The ATLL LME demonstrates reduced cytotoxic CD8^+^ T cells and B cells, impaired NK-cell function, and expansion of myeloid populations [169]. Infiltration by lymphoma-associated macrophages (LAMs) supports malignant cell proliferation, invasion, angiogenesis, and immunosuppression, particularly in acute and lymphoma-type ATLL, correlating with poor prognosis [175]. CD47 expression on ATLL cells inhibits LAM phagocytosis via SIRPα signaling (“don’t eat me” signal), though its prognostic significance remains unclear; paradoxically, SIRPα expression on stromal cells is associated with favorable outcomes [176,177].

PD-L1 expression exhibits compartment-specific effects: PD-L1 on ATLL cells, particularly in nodal lesions, is linked to poor prognosis, whereas PD-L1 on stromal macrophages and dendritic cells correlates with improved outcomes, although its precise role in the LME remains undefined [178]. Peripheral blood analyses reveal decreased invariant NK-T, NK, and dendritic cell populations in ATLL patients [179], while single-cell studies show reduced B cells, increased myeloid cells, dendritic cells, and atypical monocytes with upregulated activation markers (CD64) and immune checkpoint molecules (PD-1) [180]. Despite B-cell reduction, interferon signaling is enhanced in myeloid cells, and cytotoxic CD8^+^ T-cell decline is associated with PD-L1 genetic alterations, which further modulate immune microenvironment composition [180]. Functional NK-cell defects are observed in both HTLV-1 carriers and ATLL patients, underscoring the critical role of immune microenvironment remodeling in HTLV-1–driven carcinogenesis [169,180].

## 6. Inflammation as a Significant Factor in T/NK-Cell Neoplasms

Inflammation in mature T/NK-cell neoplasms is not a secondary consequence of malignant transformation but a fundamental driver of disease pathogenesis. The WHO-HAEM5 and ICC 2022 classifications highlight the biological distinctiveness of EBV^+^ NK/T-cell lymphomas and HTLV-1–associated ATLL, both shaped by chronic antigenic stimulation [4,5]. Malignant cells exploit the inflammatory cytokine milieu—particularly IL-15, IL-6, and IL-21—to sustain proliferation and evade apoptosis within a deregulated microenvironment [181,182].

Key inflammatory pathways, including JAK/STAT and NF-κB, are activated either by somatic mutations (*JAK3*, *STAT3*, *STAT5B*) or by viral proteins. LMP1, encoded by EBV, induces PD-L1 expression through both JAK/STAT and NF-κB signaling, whereas the HTLV-1 Tax protein serves as a potent NF-κB activator in ATLL [8,9,183]. The immune microenvironment, actively remodeled by these interactions, fosters tumor progression via immune checkpoint upregulation and inflammatory cell recruitment [182].

Clinically, this inflammatory phenotype often manifests as hemophagocytic lymphohistiocytosis–like episodes, hyperferritinemia, and elevated soluble IL-2 receptor levels, particularly in EBV-driven diseases [184,185]. Collectively, the convergence of inflammatory mediators, viral oncogenic factors, and microenvironmental reprogramming defines a recurrent pathogenic theme in T/NK-cell malignancies, positioning the JAK/STAT and PD-L1 pathways as promising therapeutic targets [4,5,7,56,83,84,85,181,182,183,184,185].

### 6.1. Leukemic Entities

In NK-LGL, the bone marrow microenvironment exhibits a highly inflamed phenotype characterized by close interactions between DCs and NK-cells, which underpin persistent antigen presentation [186,187]. Cytokines play a dual role in this setting—not only sustaining inflammation but also supporting leukemic cell survival. For instance, IL-15, secreted by stromal and DCs, promotes NK-cell persistence, while IL-6 and TNF-α produced by leukemic cells reinforce JAK/STAT and NF-κB signaling [188,189]. Moreover, IFN-γ and TNF-α can induce apoptosis of hematopoietic progenitors, contributing to cytopenias such as neutropenia, further exacerbated by IL-8 and IL-10 [13,14,15]. The hyperactive immune milieu frequently manifests with autoimmune phenomena, including rheumatoid arthritis and pure red cell aplasia, strengthening the link between chronic inflammation and leukemogenesis [187,188,189].

T-LGL similarly arises in a state of sustained inflammation, wherein immune cells secrete IL-15, IL-2, and IL-6, perpetuating malignant T-LGL survival through continuous STAT3/5 and NF-κB activation [190,191]. IL-15 and its receptor complex are markedly upregulated, while JAK/STAT signaling is amplified via overexpression of IL-2R [190]. Concurrently, the PI3K/Akt/MAPK cascade is activated through autocrine IL-6/gp130 and PDGF signaling loops, and anti-apoptotic mediators such as c-FLIP inhibit Fas-induced cell death [191]. This cytokine-driven inflammatory network underlies the anemia, neutropenia, and autoimmune manifestations characteristic of T-LGL, reinforcing the notion that chronic inflammation sustains leukemic persistence [187,192].

A unifying hallmark across mature T/NK-cell neoplasms is their capacity for self-sustaining proliferation coupled with immune evasion, both orchestrated through inflammatory mechanisms. In T-PLL, dysregulation of the IL2RG–JAK1/3–STAT5B axis results in constitutive STAT activation, generating a permissive environment for clonal expansion and survival within an immunologically imbalanced milieu [193,194].

In ANKL, a prototypical EBV-associated malignancy, patients frequently develop hemophagocytic lymphohistiocytosis (HLH) with profound macrophage activation, hyperferritinemia, and consumptive coagulopathy. This cytokine storm not only causes tissue injury but also supports the proliferation of leukemic NK-cells [121,194,195]. Immune escape in ANKL is further promoted by PD-L1 upregulation mediated through NF-κB and STAT3 activation [194,196,197].

Finally, in ATLL viral oncoproteins Tax and HBZ activate the NF-κB and AP-1 pathways, profoundly altering host immune regulation. Elevated IL-2 and IL-15 levels foster a pro-inflammatory microenvironment, whereas increased IL-10 drives a STAT3/IRF4-dependent positive feedback loop in leukemic T-cells, enhancing proliferation and immune evasion [198,199].

### 6.2. Nodal Lymphomas

Nodal T_FH_-cell lymphomas share a characteristic inflammatory microenvironment, where immune dysregulation drives both local and systemic inflammatory symptoms [200,201]. Affected lymph nodes display dense infiltrates of T cells, plasma cells, dendritic cells (DCs), and macrophages, supported by extensive follicular DC and endothelial venule networks [200]. Neoplastic TFH cells secrete IL-21 and CXCL13, stimulating germinal-center B-cell and follicular DC expansion, leading to polyclonal plasmacytosis, hypergammaglobulinemia, and Treg depletion [200,201]. IL-6 provides additional pro-survival signaling, while the accompanying cytokine storm manifests clinically with fever, rash, and weight loss [202].

PTCL-NOS and ALK^+^/ALK^−^ ALCL exhibit highly reactive microenvironments dominated by LAMs, which support tumor growth and suppress cytotoxic responses [203,204]. IL-10 activates STAT3 and inhibits DC maturation, while TGF-β reinforces immunosuppression [203,205]. In ALK^+^ ALCL, the nucleophosmin (NPM)-ALK fusion constitutively activates STAT3, inducing IL-6, IL-10, and TGF-β secretion and promoting PD-L1 expression [32,33,34,206,207,208]. In ALK^−^ ALCL, JAK1/STAT3 mutations drive similar effects, with PD-L1 expression also present on LAMs [34]. CD30-TRAF-NF-κB signaling and IRF4 cooperatively sustain cytokine production and tumor persistence [209].

PTCL-NOS comprises distinct molecular subgroups: GATA3-driven cases produce IL-4, IL-5, IL-13, and IL-10, with eosinophil/macrophage-rich infiltrates and poorer outcomes, whereas TBX21-high cases, marked by abundant IFN-γ, are associated with a more favorable prognosis [30,36,202,204].

### 6.3. Extranodal Lymphomas

Primary cutaneous acral CD8^+^ lymphoproliferative disorder is an indolent condition characterized by a localized proliferation of CD8^+^ T cells, typically affecting acral skin sites [210]. Histology reveals dense dermal infiltrates enriched in dendritic cells (DCs), macrophages, plasma cells, and eosinophils [38]. CD8^+^ T cells secrete IFN-γ and TNF-α, activating macrophages, while IL-2 supports local proliferation [211]. Macrophage-derived IL-10 suppresses DC activity, fostering immune tolerance and explaining the disease’s indolent, skin-limited course and lack of EBV association [210,211,212,213].

Intestinal T- and NK-cell lymphoid proliferations and lymphomas encompass entities driven by inflammatory mechanisms and mucosal immune activation [214]. EATL is closely linked to celiac disease, where gluten-induced IL-15 elevation, Treg suppression, and IFN-γ–mediated mucosal injury drive JAK/STAT activation [214,215]. MEITL, in contrast, arises de novo, features monomorphic epitheliotropism with minimal inflammation, and may exhibit necrosis [216]. Indolent intestinal T-cell proliferations occur in the context of IBD or immunosuppression, while their NK counterpart, indolent NK-cell enteropathy, remains EBV-negative, non-destructive, and mucosa-confined [10,217]. Shared mechanisms—IL-2, IL-15, TNF, and IFN-γ signaling—promote immune dysregulation and clonal persistence [10,214,215].

Indolent T-cell lymphoma of the gastrointestinal tract similarly shows a slow course with prominent mucosal inflammation. IBD and immunosuppression sustain small CD4^+^/CD8^+^ T-cell expansion via IL-15 and IFN-γ signaling, supporting chronic proliferation without systemic spread [218].

ENKTCL, by contrast, is an aggressive EBV-driven neoplasm marked by a cytokine-rich microenvironment and chronic inflammation [141]. Elevated serum IL-6, IL-10, survivin, and VEGF correlate with poor prognosis [219,220,221]. EBV induces a cytokine storm dominated by IFN-γ, IL-6, and IL-10, fostering systemic inflammation and PD-L1 expression that enhances immune evasion. A pathogenic feedback loop between viral activity and inflammatory mediators underlies ENKTCL progression [85,222].

### 6.4. Provisional/Childhood EBV-Driven Entities

Among EBV-driven malignancies, hydroa vacciniforme–like lymphoproliferative disorder (HV-LPD) represents a prototypical example [4,5]. Lesions contain EBV-infected CD8^+^ T and NK-cells, as well as type I and γδ T-cells [223]. The microenvironment is highly inflamed, with elevated IFN-γ and TNF-α driving keratinocyte apoptosis and recruitment of lymphocytes via chemokines CXCL9 and CXCL10 [223]. High IL-10 levels, secreted by macrophages and infected cells, confer local immunosuppression that counterbalances cytotoxic activity. Activation of the JAK/STAT and NF-κB pathways within this pro- and anti-inflammatory equilibrium underlies the chronic course of HV-LPD and the long-term survival of EBV-infected cells.

Systemic chronic active EBV disease (CAEBV) is another EBV-driven T/NK-cell disorder classified under EBV^+^ T/NK-cell lymphoid proliferations in WHO-HAEM5 [4]. It primarily involves the liver and lymph nodes, where EBV-infected cells form granulomas and portal infiltrates, resulting in chronic immune activation [224]. Elevated IL-6, TNF-α, and IFN-γ drive systemic inflammation and hemophagocytic episodes, while IL-10 promotes immune evasion and EBV persistence [225,226]. Upregulation of PD-1 and PD-L1 further contributes to immune escape [224]. Clinically, cytokine storm–mediated manifestations include cytopenia, hepatosplenomegaly, hyperferritinemia, and increased soluble IL-2 receptor levels [226,227].

## 7. Conclusions

Based on this comprehensive overview, a substantial paradigm shift is emerging in our understanding of neoplastic tissue homeostasis, particularly within the context of T/NK-cell neoplasms. The accumulating evidence indicates that these malignancies should no longer be regarded merely as aggregates of neoplastic lymphoid cells, but rather as complex, multicellular ecosystems composed of a heterogeneous admixture of stromal and immune elements intimately interwoven with lymphoma cells and the extracellular matrix. Accordingly, T/NK-cell neoplasms ought to be conceptualized as dynamic, interactive, and adaptable biological systems, responsive to a spectrum of endogenous and exogenous cues that collectively shape their behavior and evolution.

This reconceptualization provides critical insight into the pronounced clinical heterogeneity observed among T/NK-cell lymphomas, underscoring the pivotal role of LME composition in determining disease course and therapeutic response. Current data suggest that lymphomas characterized by an immune-rich stroma and elevated PD-L1/PD-1 expression are more likely to benefit from immune checkpoint inhibition (ICI), whereas those exhibiting an immune-desert phenotype may instead require therapeutic strategies directed against specific oncogenic drivers.

Advances in matrix biology are expected to catalyze the development of an expanding repertoire of targeted anti-lymphoma therapies. Beyond conventional cytotoxic regimens, the emerging vision encompasses the selective modulation of multiple molecular and cellular pathways, representing a transformative step toward personalized treatment paradigms in T/NK-cell lymphomas. While considerable progress has been made in elucidating the molecular biology of these tumors, the field remains unbalanced, and additional time and research are needed for full clinical applicability. Ultimately, the integration of high-quality, evidence-based molecular and microenvironmental markers into standard prognostic models and therapeutic algorithms promises to establish them as robust and clinically actionable tools in routine practice [228].

## Figures and Tables

**Figure 1 ijms-26-11225-f001:**
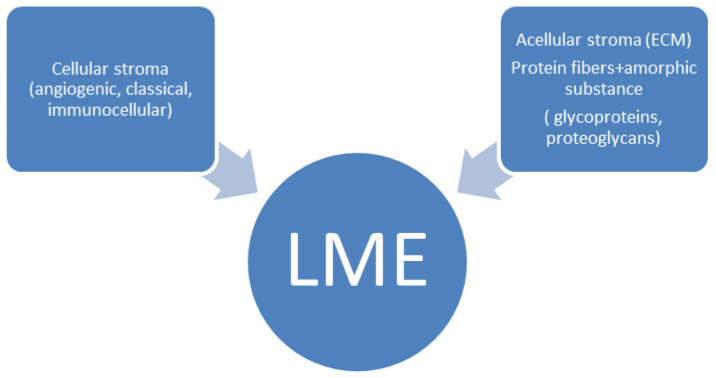
Composition of the lymphoma microenvironment.

**Table 1 ijms-26-11225-t001:** T/NK-cell neoplasms/disorders comparative table (WHO 2022 vs. ICC 2022).

WHO, 5th Edition (WHO-HAEM5) 2022	International Consensus Classification (ICC) 2022
**Mature T-cell and NK-cell leukemias (primary leukemias)**	
T-prolymphocytic leukemia (T-PLL)	T-cell prolymphocytic leukemia (T-PLL)
T-cell large granular lymphocytic leukemia (T-LGL)	T-cell large granular lymphocytic leukemia (T-LGL)
NK-cell large granular lymphocytic leukemia (NK-LGL)	Chronic LPD of NK-cells
Adult T-cell leukemia/lymphoma (ATLL)	Adult T-cell leukemia/lymphoma (ATLL)
Sézary syndrome (SSy)	Sézary syndrome (SSy)
Aggressive NK-cell leukemia (ANKL)	Aggressive NK-cell leukemia (ANKL)
**Primary cutaneous T-cell lymphomas (CTCL)**	
Primary cutaneous CD4^+^ small or medium T-cell LPD	Primary cutaneous small/medium CD4^+^ T-cell LPD
Primary cutaneous acral CD8^+^ LPD	Primary cutaneous acral CD8^+^ LPD
Mycosis fungoides (MF)	Mycosis fungoides (MF)
Primary cutaneous CD30^+^ T-cell LPD: Lymphomatoid papulosis (LyP)	Primary cutaneous CD30^+^ T-cell LPD: Lymphomatoid papulosis (LyP)
Primary cutaneous CD30^+^ T-cell LPD: Primary cutaneous anaplastic large cell lymphoma	Primary cutaneous CD30^+^ T-cell LPD: Primary cutaneous anaplastic large cell lymphoma
Subcutaneous panniculitis-like T-cell lymphoma	Subcutaneous panniculitis-like T-cell lymphoma
Primary cutaneous γ/δ T-cell lymphoma	Primary cutaneous γ/δ T-cell lymphoma
Primary cutaneous CD8^+^ aggressive epidermotropic cytotoxic T-cell lymphoma	Primary cutaneous CD8^+^ aggressive epidermotropic cytotoxic T-cell lymphoma
Primary cutaneous peripheral T-cell lymphoma, NOS	Not included
**Intestinal T-cell and NK-cell lymphoid proliferations and lymphomas (extranodal)**	
Indolent T-cell lymphoma of the gastrointestinal tract	Indolent clonal T-cell LPD of the gastrointestinal tract
Enteropathy-associated T-cell lymphoma (EATL)	Enteropathy-associated T-cell lymphoma (EATL)
Monomorphic epitheliotropic intestinal T-cell lymphoma (MEITL)	Monomorphic epitheliotropic intestinal T-cell lymphoma (MEITL)
Intestinal T-cell lymphoma, NOS	Intestinal T-cell lymphoma, NOS
Hepatosplenic T-cell lymphoma (extranodal)	
Hepatosplenic T-cell lymphoma (HSTCL)	Hepatosplenic T-cell lymphoma (HSTCL)
**Anaplastic large cell lymphoma (nodal)**	
ALK^+^ anaplastic large cell lymphoma (ALCL, ALK^+^)	Anaplastic large cell lymphoma, ALK^+^ (ALCL, ALK^+^)
ALK^-^ anaplastic large cell lymphoma (ALCL, ALK^-^)	Anaplastic large cell lymphoma, ALK^-^ (ALCL, ALK^-^)
Breast implant-associated anaplastic large cell lymphoma (BIA-ALCL)	Breast implant-associated anaplastic large cell lymphoma (BIA-ALCL)
Nodal T-follicular helper (T_FH_) cell lymphoma	
Nodal T_FH_ cell lymphoma, angioimmunoblastic-type (AITL)	T_FH_-cell lymphoma, angioimmunoblastic type (AITL)
Nodal T_FH_ cell lymphoma, follicular-type	T_FH_-cell lymphoma, follicular type
Nodal T_FH_ cell lymphoma, NOS	T_FH_-cell lymphoma, NOS
Other peripheral T-cell lymphomas	
Peripheral T-cell lymphoma, NOS (PTCL-NOS)	Peripheral T-cell lymphoma, NOS (PTCL-NOS)
EBV-positive NK/T-cell lymphomas	
EBV^+^ nodal T or NK-cell lymphoma	Primary nodal EBV^+^T/NK-cell lymphoma
Extranodal NK/T-cell lymphoma (ENKTCL)	Extranodal NK/T-cell lymphoma, nasal type (ENKTCL)
**EBV-positive T and NK-cell lymphoid proliferations and lymphomas of childhood**	
Severe mosquito bite allergy	Severe mosquito bite allergy
Hydroa vacciniforme LPD	Hydroa vacciniforme LPD
Systemic chronic active EBV disease	Chronic active EBV disease, systemic (T or NK-cell phenotype)
Systemic EBV^+^ T-cell lymphoma of childhood	Systemic EBV^+^ T-cell lymphoma of childhood

LPD—lymphoproliferative disease; NOS—not otherwise specified; ALK—anaplastic lymphoma kinase.

**Table 2 ijms-26-11225-t002:** The matrisome composition.

Class	Representative Molecules
Protein fibers	Collagens, Elastin
Glycoproteins	Fibronectin, Laminin, Tenascin
Proteoglycans	Serglycin, GAGs, Syndecan, Agrecan
Affiliated proteins	Annexins, Hemopexin, Galectins
ECM regulators	MMPs, TIMPs, Cathepsins, Serpins
Secreted proteins	Growth factors, Cytokines

**Table 3 ijms-26-11225-t003:** T/NK-cell neoplasms/disorders composition overview.

WHO-HAEM5	Cell of Origin	Immunophenotype	Specific LME
**NODAL**			
**PTCL-NOS**	Variabile, mostly T-helper cell	CD4 > CD8, frequent antigen loss CD5, CD7, CD30^+/−^, CD56^−/+^, subset F_TH_ features, cytotoxic granules^+/−^	Classical stromal cells but highly heterogenous composition
**ALCL, ALK^−^**	Cytotoxic T-cell	ALK^−^, CD30^+^, EMA^+^, CD25^+^, cytotoxic granules^+^, CD4^+/−^, CD3^+/−^	Reactive histiocytes, fibrosis, immune evasion via PD-L1
**ALCL, ALK^+^**	Cytotoxic T-cell	ALK^+^, CD30^+^, EMA^+^, CD25^+^, cytotoxic granules^+^, CD4^+/−^, CD3^+/−^	Inflammatory background, activated TME
**BIA-ALCL**	Undefined, Suggested T-cell of Th17/Th1 immune response	CD30^+^, ALK^−^, EMA^+/−^, variable T-cell markers	Fibrous capsule-related microenvironment, Th17 cytokines
**AITL**	Follicular helper T-cell (T_FH_)	Pan T^+^, CD4^+^, CD10^+/−^, bcl6^+/−^, CXCL13^+^, PD1^+^, ICOS^+/−^, SAP^+/−^, CCD5^+/−^, hyperplastic FDRC, EBV^+^ B blasts	Prominent angiogenesis (HEV), proliferating stromal cells, EBV^+^ B cells
**Nodal T_FH_ follicular-type**		Similar to AITL	Expanded FDC meshworks, angiogenesis
**Nodal T_FH_ NOS**		Similar to AITL	Variable stromal and immune infiltration
**Other PTCLs**	Variable	Depends on subtype	Heterogeneous
**EBV^+^ nodal T/NK-cell lymphoma**	Cytotoxic T or NK	EBV^+^, cytotoxic phenotype	EBV-driven microenvironment with high immune infiltration
**EXTRANODAL**			
**ENKTCL**	NK, rarely cytotoxic T-cells	CD2^+^, CD56^+^, sCD3^−^, cCD3ε^+^, granzyme B^+^, TIA-1^+^, perforin^+^, EBV^+^, LMP1^+^	Highly immune stroma, angiodestructive, necrosis
**HSTL**	Cytotoxic T-cell of the innate immune system	CD3^+^, CD56^+^, CD4^−^, CD8^+^, CD5^−^, TIA1^+^, granzyme M^+^, B^−^, perforin	Low level of stromal cells
**MEITL**	Intraepithelial T cells or NK, monomorphic, no preexisting enteropathy	CD3^+^, CD7^+^, CD5^−^, CD8^+^, CD56^+^, MATK^+^, HLA DQ2/DQ8	Sparse stroma, epithelial interaction
**EATL**	Intraepithelial T cells (αβ), pleomorphic, preexisting enteropathy	CD3^+^, CD7^+^, CD5^−^, CD8^−/+^, CD56^−^, HLA DQ2/DQ8	Inflammatory background, enteropathy-related
**LEUKEMIC**			
**T-PLL**	Post-thymic T-cell	TdT^−^, CD1a^−^, CD2^+^, CD3^+^, CD7^+^, sCD3 week, CD52^+^, CD4/CD8 variable	Leukemic spread, minimal LME
**T-LGL**	Cytotoxic T-cell	CD3^+^, CD8^+^, CD57^+^, TIA1^+^	Reactive marrow environment, immune dysregulation
**NK-LGL**	Cytotoxic NK-cell	CD2^+^, CD3^−^, CD56^+^, CD57^+^, CD16^+^	Reactive marrow niche
**ATLL**	Peripheral CD4^+^ reg cells	Pan-T-cell, CD4^+^, CD25^+^, CD7^−^	HTLV-1 driven microenvironment, immune suppression
**SSy**	Central memory T-cell CD4^+^	TH2 cytokine profile expression, CCR7/L-selectin^+^, CD27^+^, CD3^+^, CD4^+^, CD8^−^, CD7^−^, PD1^+^, bcl6^+^, CXCL13^+^	Skin homing microenvironment, immune suppression
**ANKL**	Mature, cytotoxic NK-cells	CD2^+^, CD3^−^, CD3ε^+^, CD56^+^, CD57^−^, cytotoxic phenotype, CD16^+^ frequently, CD11b expressed in some cases	Hemophagocytic environment, cytokine storm prone
**CUTANEOUS**			
**Primary cutaneous CD4^+^ small or medium T-cell LPB**	CD4^+^ T-cell	CD3^+^, CD4^+^, CD8^−^, CD30^−^	Reactive infiltrate, low stromal response
**Primary cutaneous acral CD8^+^ LPD**	CD8^+^ T-cell	CD3^+^, CD8^+^, CD4^−^, CD30^−^	Indolent, localized stroma
**Mycosis fungoides (MF)**	Mature CD4^+^ T-cell	CD2^+^, CD3^+^, CD4^+^, CD45RO^+^, TCRβ^+^, CD5^+/−^, CD7^−^, CD8^−^, CD45RA^−^, CD45 variable CD30	Skin microenvironment with Langerhans cells, fibroblasts
**Lymphomatoid papulosis (LyP) type A, B, C, D, E, w/6p25**	Activated T-cell (Reed-Sternberg like, cerbriform)	CD30^+^, CD4^+^, CD8^−/+^, variable	Inflammatory infiltrate, spontaneous regression
**Primary cutaneous ALCL**	Cytotoxic T-cell	CD30^+^, ALK^−^	Dense dermal infiltrate, reactive stroma
**Subcutaneous panniculitis-like T-cell lymphoma**	Cytotoxic T-cell (αβ)	CD3^+^, CD8^+^, TIA1^+^/βF^+^, granzyme B^+^, Ki67 elevated	Adipocyte-rich environment, macrophage infiltration
**Primary cutaneous γ/δ T-cell lymphoma**	γ/δ T-cell	CD3^+^, CD4^−^, CD8^−^, CD56^+^, TIA-1^+^, TCRγ^+^, Ki67 elevated	Ulcerating lesions, inflammatory milieu
**Primary cutaneous CD8^+^ aggressive epidermotropic cytotoxic T-cell lymphoma**	Cytotoxic CD8^+^ T-cell	CD3^+^, CD7^+^, CD8^+^, TIA-1^+^, CD45RA^+^, βF-1^+^, CD45RO^−^, CD56^−^, CD4^−^, EBER^−^	Epidermal infiltration, inflammatory background
**Primary cutaneous peripheral T-cell lymphoma, NOS**	Variable T-cell	Heterogeneous phenotype	Aggressive, heterogeneous stroma

## Data Availability

No new data were created or analyzed in this study. Data sharing is not applicable to this article.

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
