# Peer review of "The Significance of the Microenvironment in T/Nk-Cell Neoplasms"

_ijms, 2025, doi:10.3390/ijms262211225_

Round 1
Reviewer 1 Report
Comments and Suggestions for Authors
The manuscript by Petkovic and colleagues provides an extensive overview of the microenvironment in T/NK-cell neoplasms. While the topic is undoubtedly relevant, the manuscript as it stands suffers from significant issues that must be addressed to make it suitable for publication.
First and foremost, the length of the manuscript is a major concern. The review is excessively long, with much of the content reiterating basic concepts that are already well established in the literature (if not in other parts of the manuscript itself). Given the intended scope of the review, which should focus on recent advancements in the field, the manuscript would benefit greatly from significant shortening. The authors should aim for a concise and well-organized structure that highlights novel findings, rather than a detailed recapitulation of every aspect of the microenvironment. It is important to remember that this is a review, not a book chapter, and it should provide a focused synthesis of recent developments.
Additionally, many of the phrases throughout the manuscript are unnecessarily long and convoluted. In several instances, the writing is laborious and difficult to follow, which detracts from the overall clarity of the review. It would be highly beneficial for the authors to consider revising the manuscript with the assistance of a native English speaker to ensure the language is more fluid and accessible. Clear, direct writing will make the review more engaging and easier for readers to follow, ultimately improving its impact.
Furthermore, “Picture 1” doesn’t convey any relevant message. I would advise the authors to generate a drawing depicting the different LME components (including ECM, cellular stroma, regulatory factors, etc) . Moreover, some acronyms are not specified in the text before they appear (for instance, what is ICH in table 3? Do the authors mean ImmunoHistoChemistry, IHC?)
In conclusion, while this manuscript addresses a relevant and timely topic, significant revisions are necessary for it to be suitable for publication. I recommend that the authors streamline the content, reduce repetition, and revise the writing for clarity and conciseness. Once these changes are made, this manuscript may be re-assessed.
Author Response
1. First and foremost, the length of the manuscript is a major concern. The review is excessively long, with much of the content reiterating basic concepts that are already well established in the literature (if not in other parts of the manuscript itself). Given the intended scope of the review, which should focus on recent advancements in the field, the manuscript would benefit greatly from significant shortening. The authors should aim for a concise and well-organized structure that highlights novel findings, rather than a detailed recapitulation of every aspect of the microenvironment. It is important to remember that this is a review, not a book chapter, and it should provide a focused synthesis of recent developments.
Additionally, many of the phrases throughout the manuscript are unnecessarily long and convoluted. In several instances, the writing is laborious and difficult to follow, which detracts from the overall clarity of the review. It would be highly beneficial for the authors to consider revising the manuscript with the assistance of a native English speaker to ensure the language is more fluid and accessible. Clear, direct writing will make the review more engaging and easier for readers to follow, ultimately improving its impact.
Thank you for pointing this out. I agree with your observation, so I have made a significant shortening of the text in order to make it more concise, with no repetition of the same or similar phrases throughout the text. I have made English editing of the whole paper, and I hope that the presented version is now more suitable for publication (if accepted).
Regarding novelties, I think that we have made a comprehensive exploration of databases, but a small recapitulation is warranted from our point of view, because not all readers will be familiar with, for instance, extracellular matrix composition, especially clinicians. From that point, I think that most of us have forgotten many of the basic histology characteristics during clinical work, which is mostly oriented toward new agents and clinical studies. Nevertheless, we have made a significant reduction in the burden of information about basic notification regarding ECM and its compounds (glycoproteins, fibers, or proteoglycans).
2. Furthermore, “Picture 1” doesn’t convey any relevant message. I would advise the authors to generate a drawing depicting the different LME components (including ECM, cellular stroma, regulatory factors, etc).
Thank you for your observation. Picture 1 has been removed from the text because I have a similar one after this, which is now Figure 1.
I tried to find figures on T/NK-cell LME online, but didn't find anything that covers these neoplasms in full. There were a few figures on specific subtypes, such as ATLL, AITL, and ENKTCL, but none that relate to the unified presentation.
3. Moreover, some acronyms are not specified in the text before they appear (for instance, what is ICH in table 3? Do the authors mean ImmunoHistoChemistry, IHC?)
Thank you for pointing this out. I have implemented immunophenotype, that’s what I/we meant, but I did not even notice it in the table, because we routinely use ICH in our country for immunophenotype.
5. In conclusion, while this manuscript addresses a relevant and timely topic, significant revisions are necessary for it to be suitable for publication. I recommend that the authors streamline the content, reduce repetition, and revise the writing for clarity and conciseness. Once these changes are made, this manuscript may be re-assessed.
Thank you again for your kind recommendations. I think I have answered this issue in my previous discussion. The text has been shortened without significantly losing the main message from the initial content.
Reviewer 2 Report
Comments and Suggestions for Authors
The authors provide an extensive amount of information, with over 200 references, on the subject of the microenvironment in mature T/NK neoplasms. This review will potentially serve as a useful reference. Given the heterogeneity of the subject, the use of Tables is helpful. The focus on the altered immunology of the microenvironment is especially strong. Attention to the following points is necessary to improve the manuscript:
Major criticisms:
1) The quality of the English writing is suboptimal. This is manifest by the use of descriptions that are not idiomatic, or perhaps represent the choice of a style that is more poetic or emotional than is usually found in scientific writing. A particular example is this sentence in the Abstract, which is echoed in the main text: “The contents of neoplastic tissue mutually interact, displaying a characteristic of an intelligent endogenous ecosystem with the ability to harmonize its functions regarding continuous affection of multiple exogenous or endogenous factors.” One presumes that cancers are incapable of having intelligence or affection. I know that the authors are trying to say that the constituents of T/NK neoplasms, including all cell types and other microenvironmental elements, interact in a synergistic way. However, if this needs to be said (which it may not, given that it is widely recognized that tumors are pathologic neo-organs that act to their benefit), it should be said in a more reserved way that is standard for a scientific article. The manuscript needs to be edited throughout along these lines, so that the reader is not distracted from the substance of the authors’ presentation.
2) For inclusion in the Conclusion, the authors should acknowledge that while in some respects a great deal is known about the microenvironment in mature T/NK neoplasms, it is imbalanced and has yet to inform effective therapy for these intractable neoplasms. A great deal of the information presented in the review concerns ENKTL, which is a rare entity within a group of diseases that are uncommon overall. Less is known about the most common entities. Overall, mature T/NK neoplasms need more study with powerful new methods, such as ones that are revealing the microenvironment of diffuse large B-cell lymphoma (PMID: 40920660, PMID: 41120574).
Minor criticisms:
1) In line 91, the authors write that the lymphoma microenvironment is an “endogenous ecosystem composed of non-cancerous cellular (stromal) components, extracellular matrix, and a milieu of cytokines, hormones, and exosomes.” Among non-cancerous cellular components found in tumors, a distinction is usually made between cells that are fixed elements of the tissue (such as fibroblasts and endothelial cells) and those that are hematopoietic cells that have infiltrated the tumor (such as non-neoplastic lymphoid cells and myeloid cells). Both types of cells are present in T/NK neoplasms, to varying degrees that are sometimes characteristic of the particular tumor type. For example, endothelial cells and plasma cells are prominent in AITL tumors. The authors appear to recognize this, from the more detailed description provided in sections starting with lines 112 and 157. However, they should modify the earlier section to be consistent.
2) In Table 3, the column labeled “ICH” needs a better title. Presumably the authors intended to mean “IHC”, which would stand for immunohistochemistry, but a better title would be “Immunophenotype” or something similar.
3) The acronym “LAM” is first used on line 173, before it has been defined on line 265. It needs to be defined at its first use.
4) Line 348 states “high expression of IL-10 and low expression of IL-”. The second interleukin needs to be specified.
5) Line 410 has a nonsense insertion of ‘’ cioccarelil”.
Comments on the Quality of English LanguageSee above
Author Response
1. The quality of the English writing is suboptimal. This is manifest by the use of descriptions that are not idiomatic, or perhaps represent the choice of a style that is more poetic or emotional than is usually found in scientific writing. A particular example is this sentence in the Abstract, which is echoed in the main text: “The contents of neoplastic tissue mutually interact, displaying a characteristic of an intelligent endogenous ecosystem with the ability to harmonize its functions regarding continuous affection of multiple exogenous or endogenous factors.” One presumes that cancers are incapable of having intelligence or affection. I know that the authors are trying to say that the constituents of T/NK neoplasms, including all cell types and other microenvironmental elements, interact in a synergistic way. However, if this needs to be said (which it may not, given that it is widely recognized that tumors are pathologic neo-organs that act to their benefit), it should be said in a more reserved way that is standard for a scientific article. The manuscript needs to be edited throughout along these lines, so that the reader is not distracted from the substance of the authors’ presentation.
Thank you for your observations, which I accept as benevolent suggestions. I have significantly shortened the whole text (as the other reviewer asked me to do) and reassessed sentences. It sounds now in a much more reserved manner, more academic. However, I do like to use such poetic phrases occasionally in my articles as a result of my own style.
2. For inclusion in the Conclusion, the authors should acknowledge that while in some respects a great deal is known about the microenvironment in mature T/NK neoplasms, it is imbalanced and has yet to inform effective therapy for these intractable neoplasms. A great deal of the information presented in the review concerns ENKTL, which is a rare entity within a group of diseases that are uncommon overall. Less is known about the most common entities. Overall, mature T/NK neoplasms need more study with powerful new methods, such as ones that are revealing the microenvironment of diffuse large B-cell lymphoma (PMID: 40920660, PMID: 41120574).
Thank you for this specific observation. I have incorporated the sentence (While considerable progress has been made in elucidating the molecular biology of these tumors, the field remains unbalanced, and additional time and research are needed for full clinical applicability). Our intention was not to claim that we are very close to fully targeting the T/NK neoplasms, but rather that the new data are just a small step toward future treatments.
ENKTCL was one of the entities with the most provided descriptions and data about LME throughout the literature, which is why it is over-discussed. The field is not covered with comprehensive data about all of the entities equally. The review was supposed to be focused only on LME, not on other aspects of T/NK-cell neoplasms, so this was what we found.
MINOR CRITICISM:
1) In line 91, the authors write that the lymphoma microenvironment is an “endogenous ecosystem composed of non-cancerous cellular (stromal) components, extracellular matrix, and a milieu of cytokines, hormones, and exosomes.” Among non-cancerous cellular components found in tumors, a distinction is usually made between cells that are fixed elements of the tissue (such as fibroblasts and endothelial cells) and those that are hematopoietic cells that have infiltrated the tumor (such as non-neoplastic lymphoid cells and myeloid cells). Both types of cells are present in T/NK neoplasms, to varying degrees that are sometimes characteristic of the particular tumor type. For example, endothelial cells and plasma cells are prominent in AITL tumors. The authors appear to recognize this, from the more detailed description provided in sections starting with lines 112 and 157. However, they should modify the earlier section to be consistent.
Thank you for the observation. The text has been recomposed so it sounds completely different now. I hope this is more adequate.
2) In Table 3, the column labeled “ICH” needs a better title. Presumably the authors intended to mean “IHC”, which would stand for immunohistochemistry, but a better title would be “Immunophenotype” or something similar.
Thank you for your notification. Yes, ICH is immunohistochemistry or immunophenotype. I have provided a full word coloured in yellow. This was done because we routinely use ICH for immunochistochemistry in our country, so I put it there routinely.
3) The acronym “LAM” is first used on line 173, before it has been defined on line 265. It needs to be defined at its first use.
Thank you for your dedicated observation. This has been corrected because I made a significant shortening of the text, so it is now as it first appears; the acronym LAM has been used.
4) Line 348 states “high expression of IL-10 and low expression of IL-”. The second interleukin needs to be specified.
Thank you for pointing this out. It has been corrected, it is IL-12. I didn’t see it. I think that I accidentally erased it during the final preparation of the text.
5) Line 410 has a nonsense insertion of ‘’ cioccarelil”.
Thank you for pointing this out. This has been removed. The text is recomposed, so this word does not exist anymore.
Round 2
Reviewer 1 Report
Comments and Suggestions for Authors
The authors have made a commendable effort to address the previous concerns by shortening and reorganizing the manuscript. The revised version reads more clearly and flows better, which improves overall readability. While the review still does not break significant new ground in the field, it now provides a more focused synthesis of recent advancements related to the microenvironment in T/NK neoplasms. Given these improvements, the manuscript is acceptable for publication. Further enhancement could be achieved with ongoing attention to concise language and more compelling figures, but as it stands, the revision adequately meets the standards for this publisher.